# Metabolic Sequelae and Quality of Life in Early Post-Treatment Period in Adolescents with Hodgkin Lymphoma

**DOI:** 10.3390/jcm14020375

**Published:** 2025-01-09

**Authors:** Ines Pranjić, Sara Sila, Sara Lulić Kujundžić, Mateja Dodig, Anna Vestergaard Larsen, Izabela Kranjčec

**Affiliations:** 1Croatian Academic Centre for Applied Nutritional Science, 10000 Zagreb, Croatia; ines.pranjic123@gmail.com; 2Referral Center for Pediatric Gastroenterology and Nutrition, Children’s Hospital Zagreb, 10000 Zagreb, Croatia; annavlarsen1@gmail.com; 3Department of Oncology and Hematology, Children’s Hospital Zagreb, 10000 Zagreb, Croatia; sara.lulic@kdb.hr (S.L.K.); mateja.dodig@kdb.hr (M.D.); izabela.kranjcec@kdb.hr (I.K.)

**Keywords:** Hodgkin lymphoma, quality of life, metabolic syndrome

## Abstract

**Background/Objectives**: The long-term consequences of intensive treatment for Hodgkin lymphoma (HL), including metabolic syndrome (MetS) and cardiovascular diseases, but also deteriorated quality of life (QoL), are present in many survivors of childhood HL. **Methods**: Adolescents and young adults diagnosed with HL who continued the follow-up after successful treatment for HL were included. Anthropometric parameters, body composition, laboratory data, blood pressure values, compliance to the Mediterranean diet (MD), QoL and lifestyle habits were evaluated at the follow-up. Available data were also extracted retrospectively at the time of diagnosis. The primary objective was to determine metabolic sequelae in the early post-treatment period in adolescents treated for HL. Additionally, QoL and compliance with MD were explored, and the correlation of MetS with QoL was investigated. **Results**: Sixty percent of patients had at least one risk factor for metabolic syndrome, with obesity/abdominal obesity, high blood pressure and low HDL being most commonly observed, present in 66.7%, 44.4% and 44.4% of patients, respectively. The number of obese patients increased from 6.3% at the diagnosis to 31.3% at the follow-up. The majority of patients (53.3%) had low adherence to the MD. Participants had comparable quality-of-life domains to those of the healthy population at the follow-up. The physical health domain of QoL was positively correlated with compliance to the MD in young adults (r = 0.8, *p* = 0.032) and negatively correlated with obesity/overweight in adolescents (r = −0.85, *p* = 0.008). **Conclusions**: Healthy lifestyle choices can impact not only the metabolic health of survivors but also their quality of life, and therefore should be encouraged in these patients.

## 1. Introduction

Metabolic syndrome (MetS), generally defined as a combination of a few interconnected cardiovascular risk factors, including obesity, hypertension, dyslipidemia and insulin resistance, has become one of the major health burdens of modern civilization [1,2,3,4]. Namely, one-quarter of the world’s population is believed to suffer from this severe condition predisposing the individual to type 2 diabetes and cardiovascular disease (CVD) [5]. The most serious acute CVD events, related to significant mortality, clinically manifest as strokes and heart attacks [5].

Although a range of articles on MetS in the pediatric population have been published over the last two decades, incorporating all possible definitions including variables such as fasting glucose, waist circumference, triglycerides and high-density lipoprotein levels, as well as blood pressure measurements, no single cut-off point can be easily determined due to body changes related to growth and development specific to this age group [3]. Therefore, as a globally accepted definition of MetS in children is still lacking, the reported prevalence in the pediatric and adolescent population ranges from 0.3 to 26.4%, depending on the categorization used [6,7]. Particularly worrisome is the high and steadily rising occurrence of endocrine disturbances, mostly diabetes and MetS in adult survivors of pediatric malignancy, reaching up to 63% [8].

Moreover, childhood cancer survivors are at an increased risk of CVD unrelated to MetS, which has been attributed to treatment factors rather than genetic variations that might determine inter-individual vulnerability to MetS in the general population [9]. Brain surgery and radiation therapy, abdominal and total body irradiation with consecutive endocrine organ damage, oxidative stress and cell growth disruption caused by cytostatic agents and reduced physical activity accompanied by altered nutritional intake have all been identified as predisposing factors for CVD in pediatric patients treated for malignancy, with the presence of MetS additionally accelerating the evolution of CVD in adulthood [10,11]. Notwithstanding the enormous burden of endocrine and cardiovascular morbidity, CVD has been identified as the primary cause of early mortality in childhood cancer survivors [12].

Every component of MetS, quite understandably, appears to be even more prevalent in adult survivors of pediatric cancer [13,14,15]. Obesity is increasingly being recognized as a late effect, especially in acute lymphoblastic leukemia (ALL) patients, appearing early in the survivorship [13]. On the contrary, the prevalence of hypertension in cancer survivors compared to the general population is likewise increased but the condition develops in older age, regardless of malignancy and its specific treatment [14]. Goldberg and associates found a higher frequency of dyslipidemia, reflected as higher LDL cholesterol and lower HDL cholesterol, in a cohort of more than four thousand childhood cancer survivors compared to community controls [15].

Regardless of the previous contradictory results, quality of life (QoL) seems to be significantly impaired in adults with MetS, as reported in the systematic review by Saboya et al. [16] and confirmed in several later studies [17,18,19]. Although scarce data are available on the influence of MetS on QoL in children and adolescents, and vice versa, obesity itself negatively impacts QoL in this population group [20,21]. Elevated blood pressure seems to bear the same harmful effect on health-related QoL in the pediatric population, as recently reported by Slovenian and Ukrainian colleagues [22,23].

Hodgkin lymphoma (HL) constitutes less than 10% of childhood cancer; however, it is the predominant malignant disease in children aged 15 to 19 years, with a slight female prevalence [24]. A few risk factors for HL have so far been identified, including familial predisposition, immunodeficiency and viral exposure [24]. More than 90% of HL cases are classified as classical HL (cHL), which can be further subdivided into nodular sclerosis, mixed-cellularity, lymphocyte-rich and lymphocyte-depleted subtypes, while the nodular lymphocyte-predominant category (NLPHL) is less common [24]. Most children present with lymphadenopathy; very often, mediastinal mass is verified, and nonspecific systemic symptoms (fever, weight loss and drenching sweats) are also frequently reported [24]. A histological analysis of biopsy specimens is crucial for diagnosis, while staging according to the Ann Arbor classification relies on imaging studies, recently most commonly 18-fluoro-2-deoxyglucose (FDG)-positron emission tomography (PET) scans [24]. Treatment includes various chemotherapy risk-adapted regimens with preferable omission of radiotherapy, and over the last decade, the First international Inter-Group Study for classical Hodgkin’s Lymphoma in Children and Adolescents (EuroNet-PHL) protocols have been adopted for use by the many European pediatric oncology centers.

Since contemporary Hodgkin lymphoma treatment protocols guarantee excellent outcomes, albeit with considerable long-term late effects, focus has been set on elaborate evaluation of QoL in these patients, both during the acute management and survivorship [25]. Endocrine dysfunction, second malignancies and impaired general well-being have been listed as the frequent late sequelae of HL treatment [24]. Severely reduced QoL as a result of psychological issues, emotional distress and cognitive impairment, has been reported by some adolescents treated for HL, while in others poor QoL was related to somatic manifestations, such as intense fatigue or neuropathy-related symptoms [26,27,28]. Additional decline in already-decreased QoL later in survivorship is influenced predominately by newly developed cardiac and pulmonary manifestations [29]. Besides secondary malignancy, cardiovascular disease has been apostrophized as the leading cause of death in HL long-term survivors.

The primary objective of our study was to determine metabolic sequelae in the early post-treatment period in adolescents treated for HL. Additionally, QoL and compliance with the Mediterranean diet were explored, and the correlation of MetS with QoL was investigated.

## 2. Materials and Methods

### Study Design

The study population included adolescents diagnosed with Hodgkin lymphoma at the Department of Oncology and Hematology, Children’s Hospital Zagreb, Croatia, from 2016 to 2023. Included were all patients who continued the follow-up at the department and for whom the minimum of 6 months of post-treatment period had lapsed. Excluded were patients who missed the follow-up appointment, who finished their treatment less than 6 months prior and who did not sign the informed consent.

Informed consent was obtained from all individual participants and at least one of their parents included in the study.

Epidemiological (age, sex, year of diagnosis) and clinical data (exact histopathological diagnosis, disease stage, treatment protocol, number of chemotherapy cycles, radiotherapy, relapse, autologous hematopoietic stem cell transplant, outcome) were retrieved from the patient’s electronic medical records.

At the time of diagnosis, available data were retrieved retrospectively from patients’ electronic medical records, and these included the following:Anthropometric indices (body weight (kg), body height (cm), body mass index (BMI) (kg/m^2^));Blood pressure value (measured by electric device, mmHg);Available laboratory parameters (serum glucose (mmol/L), uric acid (mmol/L) and C-reactive protein (CRP) (mg/L)).

At a regularly scheduled follow-up visit participants were seen by the oncologist, dietitian and psychologist, and the following information was gathered:Anthropometric indices (body weight (kg), body height (cm), BMI (kg/m^2^), waist circumference (cm));Body composition (measured by bioelectric impedance);Blood pressure value (measured by electric device, mmHg);Laboratory parameters (serum glucose (mmol/L), insulin (pmol/L), cholesterol (mmol/L), high-density lipoprotein (HDL) (mmol/L), low-density lipoprotein (LDL) (mmol/L), triglycerides (mmol/L), insulin growth factor 1 (IGF-1) (µg/L), uric acid (mmol/L), CRP (mg/L), interleukin-6 (IL-6) (pg/mL), apolipoprotein B (apoB) (g/L);Compliance with the Mediterranean diet (MD) (KIDMED and MEDAS questionnaires)Lifestyle habits (using questionnaire; presence of sleep problems (yes/no), average number of hours of sleep, alcohol consumption (yes/no), smoking (yes/no), regular physical activity (yes/no));Quality of life (QoL) (WHOQOL-BREF and KIDSCREEN-27 questionnaires).

Adherence to the Mediterranean diet (MD) was determined using two questionnaires: in patients who were 18 and older, the Mediterranean Diet Adherence Screener (MEDAS) was used [30,31], and in patients younger than 18 years, the Mediterranean Diet Quality Index (KIDMED) [32,33] questionnaire was performed. Both were completed by the participants themselves. The final MEDAS score can range between 0 and 14 points, while for KIDMED, that is −4 to 12 points. The level of adherence was determined for each patient (low adherence: ≤5 for MEDAS and ≤3 for KIDMED, moderate adherence: 6–9 for MEDAS and 4–7 for KIDMED, high adherence: ≥10 for MEDAS and ≥8 for KIDMED). Both questionnaires have previously been adapted for the Croatian language and tested for reliability and validity [30,32].

The World Health Organization Quality of Life Brief Version (WHOQOL-BREF) is an abbreviated, 26-item version of the 100-item WHOQOL-100 quality-of-life measure [34]. It was developed for the assessment of the quality of life of adults, and was thus used in our study for patients aged 18 years or older. The WHOQOL-BREF addresses four quality-of-life domains: physical health, psychological health, social relationships and environment. It is also available in Croatian language [35]. From the previous studies that have used WHOQOL-BREF, the following scores were extracted and applied for the assessment of QoL: scores ≤ 45—low QOL; scores 46–65—moderate QOL; scores > 65—high QOL [36].

KIDSCREEN-27 is the intermediate version of the KIDSCREEN questionnaires with 27 items [37]. This version can be used when various aspects of health-related quality of life in children and adolescents are to be surveyed. It was developed to create a shorter version of KIDSCREEN-52. KIDSCREEN-27 measures the following five dimensions: physical well-being, psychological well-being, autonomy and parent relations, peers and social support and school environment. It is also available in Croatian language. T-score for KIDSCREEN-27 was compared to the European norm data for adolescents, available on the KIDSCREEN webpage [38].

Nutritional status was determined at the diagnosis and the follow-up. At the diagnosis, BMI and BMI SDS were determined for each participant, and nutritional status was defined as follows: underweight (BMI SDS < −1), normal weight (BMI SDS −1 to 1), overweight (BMI SDS > 1) and obesity (BMI SDS > 2). At the follow-up, nutritional status was defined as previously described for patients who were younger than 18 years old, while for those who were aged 18 and older, it was defined as follows: underweight (BMI < 18.5 kg/m^2^), normal weight (BMI 18.5–24.9 kg/m^2^), overweight (BMI 25.0–29.9 kg/m^2^) and obesity (BMI > 30 kg/m^2^). Body composition was determined using Maltron BF906 bioelectrical impedance (Maltron International Ltd., Rayleigh, Essex, UK). Maltron BF906 is a tetra-polar device, with an impedance of 200–1000 Ω, precision of ±4 Ω, and a frequency of 50 kHz. Patients were fasting for at least 4 h before testing, and the measurement was performed with the participant lying in the supine position. 

The presence of MetS was evaluated only at the follow-up. MetS was defined according to the International Diabetes Federation (IDF) consensus definition of the metabolic syndrome in children and adolescents [3]. For children aged 10 years or older, metabolic syndrome can be diagnosed with abdominal obesity and the presence of two or more other clinical features (i.e., elevated triglycerides, low HDL cholesterol, high blood pressure, increased plasma glucose). For adolescents who are aged 16 and older, the adult criteria can be used: central obesity (waist circumference with ethnicity-specific values) plus any two of the following four factors: 1. raised triglycerides, >1.7 mmol/L; 2. reduced HDL cholesterol, <1.03 mmol/L in males and 1.29 mmol/L in females; 3. raised blood pressure (systolic > 130 mmHg or diastolic > 85 mmHg); 4. raised fasting plasma glucose (>5.6 mmol/L).

This study was approved by the Ethics Committee (protocol code 02-23/17-2–23, date of approval 4 September 2023) and was performed in accordance with the 1964 Helsinki Declaration and its later amendments or comparable ethical standards.

The normality of data was assessed by the Shapiro–Wilk test. Numerical data were presented as median (IQR) and categorical data as N (%). Differences between pairs of samples were determined using related samples Wilcoxon signed rank test. Differences between male and female patients were tested using the independent samples Mann–Whitney U test. Spearman’s rank order correlation was performed to test for a correlation between adherence to the Mediterranean diet and quality-of-life domains with other variables related to metabolic syndrome. A *p*-value < 0.05 was considered statistically significant in two-sided tests.

## 3. Results

Demographic data of included patients are demonstrated in Table 1. All patients were diagnosed with cHL; in more than half of cases, histopathological type of nodular sclerosis was verified. The EuroNet-PHL-C1 chemotherapy protocol was administered as a first-line therapy in all participants. According to this protocol, three treatment groups (TG1-3) are recognized depending on the stage and presence of the systemic symptoms, two to six chemotherapy cycles are administered, and irradiation is provided for the selected cases. Glucocorticoids (prednisone, either 60 mg/2 or 40 mg/m^2^) in a 15-day course are integral components of every cycle. One patient switched to ABVD protocol (doxorubicin, vinblastine, dacarbazine, bleomycin) due to serious adverse events (SAE) during the third and fourth chemotherapy cycles related to cyclophosphamide. The IGEV (ifosfamide, gemcitabine, vinorelbine, prednisolone) protocol was a second-line therapy of choice in two out of three relapsed patients. At the same time, the third one experienced two relapses, and was treated with GDP (gemcitabine, dexamethasone, cisplatin) and BBV (bendamustin, brentuximab vedotin) cycles, respectively. BEAM (carmustine, etoposide, cytarabine, melphalan) conditioning was used in both patents in whom autologous stem cell transplant was indicated, To date, all 16 patients remain in remission.

At the follow-up, eight patients (50%) were older than 18 years.

### 3.1. Metabolic Syndrome

Overall, one patient (6.7%) fulfilled the criteria for metabolic syndrome according to the IDF criteria. A number of existing risk factors for metabolic syndrome were determined in 15/16 patients: 60.0% of patients had at least one risk factor for metabolic syndrome, and 33.3% of patients had two or more metabolic syndrome risk factors. In patients who had at least one risk factor for metabolic syndrome, obesity/abdominal obesity, high blood pressure, and low HDL were most commonly observed, present in 66.7%, 44.4% and 44.4% of patients, respectively. One patient received antihypertensive therapy.

Table 2 shows the differences in specific laboratory values and anthropometric parameters at the diagnosis (for available parameters) and at the follow-up (Table 2). There was a significant increase in BMI (kg/m^2^) and uric acid level at the follow-up compared to values at the diagnosis.

The nutritional status of all patients was evaluated at the diagnosis and the follow-up. The number of obese patients increased from 6.3% at the diagnosis to a notable 31.3% at the follow-up (Figure 1).

There were no statistically significant differences for any of the analyzed variables between boys and girls.

### 3.2. Mediterranean Diet

The majority of patients (53.3%) had low adherence to the MD, while 33.3% of patients had moderate and 13.3% had high adherence to the MD. There was a significant negative correlation between adherence to the MD and body fat percentage (r = −0.83, *p* = 0.042). No statistically significant correlations were found for other analyzed variables.

### 3.3. Quality of Life

A separate analysis for quality of life was performed for patients younger than 18 years old (Table 3) and for those 18 and older (Table 4). As shown in Table 3, adolescent patients had comparable quality-of-life domains to healthy children. Similarly, young adult patients had high scores for all quality-of-life domains.

### 3.4. Correlation Between QoL Domains and Other Outcomes in Adolescents (KIDSCREEN-27)

Physical well-being was negatively correlated with obesity and overweight (r = −0.85, *p* = 0.008). No correlations were detected between quality-of-life domains and other metabolic syndrome risk factors.

### 3.5. Correlation Between QoL Domains and Other Outcomes in Young Adults (WHOQOL-BREF)

Physical health and environmental health domain were positively associated with adherence to the MD (r = 0.8, *p* = 0.032 for both physical health and environmental health domains). The physical health domain was additionally negatively correlated to LDL at the follow-up (r = −0.8, *p* = 0.031). The social relationships domain was positively correlated to HDL at the follow-up (r = 0.83, *p* = 0.02), while the environmental health domain was negatively correlated to body weight z-score at the diagnosis (r = 0.73, *p* = 0.04). The psychological health domain was positively associated with hours of sleep (r = 0.8, *p* = 0.015). No correlations were detected between quality-of-life domains and other metabolic syndrome risk factors.

## 4. Discussion

Sixty percent of Hodgkin lymphoma survivors had at least one, while 33.3% of patients had two or more metabolic syndrome risk factors, most commonly obesity/abdominal obesity, high blood pressure and low HDL. Fifty percent of patients were overweight or obese at the follow-up. The majority of patients had low to moderate compliance with the MD. Quality of life does not seem to be impaired in survivors of HL.

At a median follow-up of 4.3 years after diagnosis, only one patient fulfilled criteria for metabolic syndrome. However, obesity was present in 31.25% of included patients. Similarly, in a study by Ehrhardt MJ et al. [39], obesity was present in 35% of HL survivors. It was previously shown that BMI is a strong predictor of metabolic syndrome in children [40]; therefore, overweight and obesity in our patients present an important risk factor for the future development of metabolic syndrome and cardiovascular diseases. Indeed, survivors of adolescent and young adult cancer were more likely to purchase prescription drugs for diseases in metabolic syndrome, and this was especially evident among survivors of Hodgkin lymphoma (RR, 2.40 for diabetes) [41]. Although the majority of our patients did not fulfill the criteria for metabolic syndrome, about 1/3rd of them had two or more metabolic syndrome risk factors (most commonly obesity, high blood pressure and low HDL). It is likely too early in survivorship to detect more prominent consequences on the metabolic and cardiovascular health of our patients.

The link between HL and MetS is multifactorial. However, it seems that obesity is a crucial component of metabolic syndrome. Exposure to glucocorticoids such as dexamethasone has be shown to be one of the main risk factors for obesity. Glucocorticoids regulate the maturation process of pre-adipose cells into adipose cells and promote increased fat mass, and induce impaired insulin signaling. However, the mechanism of MetS development in HL survivors is not entirely clear, and it seems that a combination of hormone deficiencies, changes in insulin sensitivity, lipid metabolism, and endothelial damage could be related to the development of MetS.

The burden of potentially modifiable cardiometabolic risk factors among childhood cancer survivors compared with population-matched controls was explored in a study by Lipshultz ER et al. [42]. Interestingly, in their study, although survivors had higher overall rates of pre-hypertension/hypertension, they had fewer metabolic syndrome features compared with healthy controls (≥2 features: 26.7% vs. 55.9%; *p* < 0.001). However, participants who accepted participation in this study were more likely to be female and White non-Hispanic, and, as indicated by the authors, it is possible that participants with healthier lifestyle profiles may have been more likely to participate in the study [42]. Indeed, cancer survivors were more physically active and smoked tobacco less as compared to healthy controls (both *p* < 0.0001). Although the results of this study are somewhat contradictory to other studies, they indicate that healthy lifestyle choices in survivors of childhood and adolescent cancer might be able to prevent the development of the metabolic syndrome and its consequences later in life. Therefore, special attention should be paid to early interventions in childhood HL survivors. Indeed, although the median age of our participants was only 17.9 years, 12.5% and 18.8% of participants stated they smoked cigarettes and consumed alcohol, respectively. Moreover, only 37.5% of participants had regular physical activity. These results presents the need for lifestyle interventions starting early in survivorship.

The role of diet in the risk of cardiovascular disease in survivors of childhood cancer was explored in previous studies [43]. Greater adherence to healthy dietary patterns was associated with a lower risk of cardiovascular diseases in childhood cancer survivors, and this was more apparent in women than in men [43]. A systematic review by Li R et al. [44] demonstrated that better adherence to healthy dietary recommendations is associated with lower CVD risk factors in childhood cancer survivors. In our cohort, more than 50% of patients had low adherence to the Mediterranean diet. Indeed, we found a significant negative correlation between adherence to the MD and body fat percentage, indicating that higher MD compliance is associated with lower body fat percentage and hence, better nutritional status. Similarly, in a study by Tonorezos ES et al. [45], greater adherence to a Mediterranean diet pattern was associated with lower visceral adiposity, subcutaneous adiposity, waist circumference, and body mass index. Moreover, the odds of having the metabolic syndrome fell by 31% for each point higher on the Mediterranean Diet Score. This study included 117 adult survivors of childhood acute lymphoblastic leukemia, with a follow up of 17.6 years after cancer diagnosis. We may speculate that the lack of detectable associations between diet and other metabolic or anthropometric outcomes, except body fat, could be due to the small sample size. Additionally, the relatively short time since cancer diagnosis may not yet allow the consequences of lifestyle choices and intensive chemotherapy to become evident.

In our cohort of patients, adolescent survivors of HL had comparable T-scores for QoL domains compared to norm data, and young adult survivors of HL scored ≥65 for WHOQOL-BREF, indicating high quality of life. Similarly, in a study by Hu Y et al. [46], authors found no significant differences compared with the general population in survivors of childhood non-Hodgkin’s lymphoma. In a study by Ayad A et al. [47], the total mean QoL score was classified as “satisfactory.” Kreissl S et al. [48] previously demonstrated that health-related QoL improved quickly after chemotherapy, which is in accordance with our results. Since we do not have data on quality of life at the diagnosis, we cannot tell whether the quality of life was affected at the start of treatment. Interestingly, physical well-being was negatively correlated with overweight/obesity in adolescent survivors, while in young adults, the physical health domain was negatively correlated to LDL at the follow-up. To our knowledge, this is the first study to explore associations of quality-of-life domains with metabolic and anthropometric outcomes in adolescent and young adult survivors of HL. Some studies indicate that healthy lifestyle choices, such as regular physical activity, are associated with better functional well-being and overall QOL in lymphoma survivors [49]. Moreover, a recent systematic review and meta-analysis demonstrated that higher MD compliance can reduce cancer patients’ weight and fat mass, improve their quality of life, and alleviate fatigue [50]. Taken together, it seems that healthy lifestyle choices, such as compliance with the MD and regular physical activity can reduce the risk of developing metabolic syndrome, and can also contribute to a better quality of life.

The main limitations of our study are the small sample size, the lack of a control group, and the retrospective evaluation of variables at the diagnosis. Moreover, not all indices of MetS were available at the time of diagnosis. Therefore, we were not able to detect changes that occurred after treatment. Nevertheless, our study adds to the existing data on metabolic syndrome, diet and quality of life in early survivorship of patients treated for Hodgkin lymphoma.

## 5. Conclusions

Our study demonstrates that obesity, high blood pressure, and low HDL are present early in the survivorship of patients treated for Hodgkin lymphoma. Interestingly, the quality of life of adolescent and young adults survivors of HL is similar to that of healthy peers and does not seem to be impaired in any of the evaluated domains, including the health domain. The physical health domain of the QoL was positively correlated to compliance with the Mediterranean diet in young adults, and negatively correlated with obesity and overweight in adolescents, indicating that healthy lifestyle choices can impact not only the metabolic health of survivors but also their quality of life. Therefore, a healthy diet, regular physical activity, and cessation of smoking and alcohol consumption should be encouraged in all patients.

## Figures and Tables

**Figure 1 jcm-14-00375-f001:**
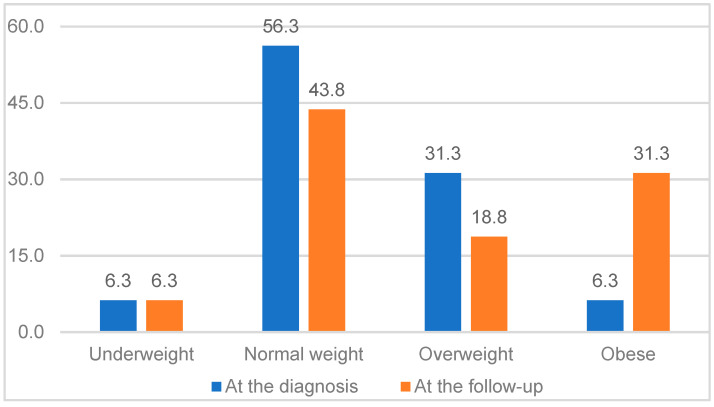
Nutritional status of patients (*n* = 16).

**Table 1 jcm-14-00375-t001:** Descriptive statistics of included patients at the follow-up.

	All Patients (*n*= 16)
Female, *n* (%)	9 (56.3)
Age, median (IQR)	17.9 (4.7)
Age at diagnosis, median (IQR)	15.0 (3.0)
Years from the diagnosis, median (IQR)	4.3 (2.4)
Radiotherapy, *n* (%)	5 (31.3)
Relapse, *n* (%)	3 (18.8)
Chemotherapy cycles (*n*), median (IQR)	6 (2)
Autologous hematopoietic stem cell transplant, *n* (%)	2 (12.5)
Body weight (kg), median (IQR)	73.0 (64.8)
Body height (cm), median (IQR)	170.0 (13.9)
BMI (kg/m^2^), median (IQR)	26.9 (11.5)
Hours of sleep (*n*), median (IQR)	7.5 (2)
Cigarette smoking, *n* (%)	2 (12.5)
Alcohol consumption, *n* (%)	3 (18.8)
Regular physical activity, *n* (%)	6 (37.5)

**Table 2 jcm-14-00375-t002:** Risk factors for metabolic syndrome and other laboratory results.

	At the Diagnosis (*n* = 16)	At the Follow-Up (*n* = 16)	*p*-Value
BMI (kg/m^2^), median (IQR)	21.5 (8.3)	26.9 (11.5)	0.001
Waist circumference (cm), median (IQR)	/	85.0 (19.1)	
Body fat (%), median (IQR)	/	31.6 (20.9)	
Systolic blood pressure (mmHg), median (IQR)	113 (25) *	121 (55)	0.091
Diastolic blood pressure, median (IQR)	70 (20) *	70 (16)	0.889
Cholesterol (mmol/L), median (IQR) (reference range < 5.0)	/	4.0 (1.6)	
HDL (mmol/L), median (IQR) (reference range > 1.2)	/	1.4 (0.5)	
LDL (mmol/L), median (IQR) (reference range < 3.0)	/	2.4 (0.9)	
Triglycerides (mmol/L), median (IQR) (reference range < 1.7)	/	0.7 (0.3)	
Serum glucose (mmol/L), median (IQR) (reference range 4.2–6.0)	5.3 (1.3)	4.6 (0.8)	0.051
Insulin (µg/L), median (IQR)	/	54.0 (56.5)	
IGF-1 (ng/mL), median (IQR)	/	332 (108)	
apoB (g/L), median (IQR)	/	0.76 (0.33)	
CRP (mg/L), median (IQR) (reference range 0–5)	21.4 (53.6)	2.0 (3.4)	0.004
IL-6 (pg/mL), median (IQR) (reference range < 7.0)	/	7.2 (5.8)	
Uric acid (µmol/L), median (IQR) (reference range 134–337)	246.5 (198)	346 (112)	0.041

* Available for 9 of the patients; BMI—body mass index; HDL—high-density lipoprotein; IQR—interquartile range; LDL—low-density lipoprotein.

**Table 3 jcm-14-00375-t003:** Quality-of-life domains (<18 years) (*n* = 8), KIDSCREEN-27.

	T-Score, Median (IQR)	European Norm data KIDSCREEN-27 for Adolescents, Mean (SD)
Physical Well-Being	44.73 (9.0)	46.83 (9.15)
Psychological Well-Being	50.8 (10.1)	47.30 (9.58)
Autonomy and Parent Relations	55.8 (11.0)	48.53 (9.75)
Peers and Social Support	51.1 (15.0)	50.07 (9.97)
School Environment	45.4 (8.14)	48.54 (9.15)

**Table 4 jcm-14-00375-t004:** Quality-of-life domains (≥18 years) (*n* = 8), WHOQOL-BREF.

	Score, Median (IQR)
Physical health	78.6 (8.9)
Psychological health	77.1 (35.4)
Social relationships	75.0 (12.5)
Environmental health	84.4 (16.4)

## Data Availability

The data presented in this study are available on request from the corresponding author due to privacy of patients involved in research.

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
