# Peer review of "Metabolic Sequelae and Quality of Life in Early Post-Treatment Period in Adolescents with Hodgkin Lymphoma"

_jcm, 2025, doi:10.3390/jcm14020375_

Round 1

Reviewer 1 Report

Comments and Suggestions for Authors   I have read and evaluated the manuscript entitled "Metabolic sequelae and quality of life in early post-treatment period in adolescents with Hodgkin lymphoma". The study is interesting, however, at this point, it requires major revisions. The authors do not mention if informed consent was sought from adult patients aged at least 18 and from the parents of the children of those aged less than 18. The study is retrospective in nature and the sample size is too low, also the follow-up is way too short. The authors did not assess MetS indices at the diagnosis of Hodgkin's lymphoma which precludes the interpretation of these results.  Also, you did not discuss the pathophysiology behind the development of MetS in Hodgkin's lymphoma> what is the link? Since you don't have reference values for the moment of diagnosis, it is rather impossible to say whether this blood cancer influenced MetS parameters. In addition, I don't see the link between the Mediterranean diet and your results since the patients did not adhere to this diet.  Maybe it would be interesting to focus just on QoL at the moment, however, a follow-up of just 6 months is not enough, you need to follow them for at least 1 year.  

Author Response

Dear Reviewer,

Thank you for giving us the opportunity to submit a revised draft of the manuscript “Metabolic sequelae and quality of life in early post-treatment period in adolescents with Hodgkin lymphoma” for publication in the Journal of Clinical Medicine. 

We appreciate the time and effort that you dedicated to providing feedback on our manuscript and are grateful for the insightful comments and valuable improvements to our paper.

We have incorporated most of the suggestions made by you and fellow reviewer, and changes are highlighted blue within the manuscript.

Please see below for a point-by-point response to your comments and concerns.

Comment 1: Comment regarding informed consent.

Response 1: Thank you for your comment. We have added this information into the manuscript more clearly.

Comment 2: Comment regarding sample size and a follow-up duration.

Response 2: Thank you for your comment. We are aware of our small sample size and a short follow-up duration. However, the goal of our study was to show that risk factors for metabolic syndrome develop early into survivorship, which is an added value of this study. Regarding the sample size, with the population of about 3 million, Croatian scientific medical community is constantly battling with ‘small sample size’ as a limitation in medical research. According to the Croatian Cancer Registry, 99 citizens were diagnosed with HL in 2021, of whom 17 aged 0-19 years, treated at four different pediatric oncology centers.

Comment 3: Comment regarding MetS indices at the diagnosis.

Response 3: Thank you for your comment. The primary objective of our study was to determine metabolic sequelae in early post-treatment period in adolescents treated for HL. At the start of the treatment, only glucose, blood pressure and BMI were evaluated. Other MetS indices such as triglycerides and cholesterol were determined in minority of patients at the time of diagnosis. This is because majority of patients had normal values of fasting glucose as well as BMI z-scores, and therefore, lipid disorders, according to our vast clinical experience, were not anticipated . Nevertheless, we are aware that this is limitation of our study. However, we’ve decided that adding available data from the start of treatment would give us a better, although not ideal, overview of our patients and changes which have occurred after the end of treatment.

Comment 4: Comment regarding the link between HL and MetS.

Response 4: Thank you for your comment. Link between HL and MetS is multifactorial. However, it seems that obesity is a crucial component in metabolic syndrome. Exposure to glucocorticoids such as dexamethasone has been shown to be one of the main risk factors for obesity. Glucocorticoids regulate the maturation process of pre-adipose cells into adipose cells and promote increased fat mass, as well as induce impaired insulin signaling. However, mechanism of MetS development in HL survivors is not entirely clear, and it seems that combination of hormone deficiencies, changes in insulin sensitivity, lipid metabolism, endothelial damage, and could be related to the development of MetS.

This has now been explained in more detail in the Discussion section of the Manuscript.

Comment 5: Comment regarding the Mediterranean diet.

Response 5: Thank you for your comment. Low adherence to the Mediterranean diet is likely, among other reasons, linked to the increased prevalence of obesity, as indicated by negative correlation between adherence to the Mediterranean diet and body fat percentage. Having in mind that abdominal obesity is the most frequently observed component of metabolic syndrome (https://pubmed.ncbi.nlm.nih.gov/28585193/), we can only assume that obese patients would be more likely to develop metabolic syndrome later in life, which was not yet evident this early in survivorship.   

Reviewer 2 Report

Comments and Suggestions for Authors

This article evaluated several metabolic syndrome variables and the quality of life of adolescents who had received treatment for Hodgkin lymphoma. The limitation is the number of cases (this is acknoledged at the end of the manuscript). Additionally, some variables were not measured before treatment. Therefore, I am not sure if "metabolic sequelae" should be used (authors know best). The article is easy to read and understand.

Additional comments:

(1) Lines 35-40. The authors could add some information (if they agree with this comment) regarding the different definitions of metabolic syndrome, which include NCEP ATP III 2005*, IDF 2009, EGIR 1999, WHO 1999, and AACE 2003.

(2) Line 42. Regarding "globally accepted definition of MetS in children is still lacking". They authors could mention the definitions of metabolic syndrome in children and adolescents, including the Modified ATP III, IDF (10 to 16 years), and NHANES III.

(3) Line 58. Regarding "Every component of the MetS per se, quite understandably, appears to be even more prevalent in adult survivors of pediatric cancer". Could you please add the reference?

(4) In the Introduction, the authors could make a brief overview of the clinicopathological characteristics of Hodgkin lymphoma in children and adolescents.

Useful sentences are as follows:

(4.1) "Hodgkin lуmрhοmа (HԼ, formerly called Hodgkin's disease) is a malignant lуmphоma that accounts for approximately 7 percent of ϲhilԁhοоd cancers and 1 percent of ϲhildhοοԁ ϲаոсer deaths in the United States"

"The incidence of НԼ in ϲhilԁhοοd varies by age such that HL is exceedingly rare in infants, but is the most common ϲhilԁhοοd ϲаnсer in the 15- to 19-year-old age group."

"The overall incidence of HL in аԁοlеѕсеոts is greater in females than in males (male to female ratio 0.8); however, boys under age 15 have higher incidence of ΗԼ, with up to fivefold higher incidence of HԼ in boys than girls less than five years of age"

(4.2) Predisposing factors:

(a) Immunodeficiency (fore example, ataxia telangiectasia, chromosomal breakage syndromes, and the autoimmune lymphoproliferative syndrome.

(b) Familiar (familial HL has been estimated to represent 4.5 percent of all cases of HL. This familial association may include shared environmental factors, exposure to viruses, and genetic influences, including inherited immunodeficiency states).

(4.3) Histologic classification

(4.4) Clinical presentation

(4.5) Evaluation, diagnosis, and differential diagnosis

(4.6) Staging

(4.7) Treatment

(5) The authors may briefly describe the late complications of the treatment. Classically, the complications were impaired grоwth of soft tissue and bones, thyroid dysfunction, gonadal dysfunction, cardiopulmonary toxicity, second malignancies, and functional impairment and reduced overall general health.

(6) What kind of equipment was used to measure the bioelectrical impedance?

(7) Line 118. Please add the website links:

https://www.who.int/tools/whoqol/whoqol-bref

https://www.kidscreen.org/english/questionnaires/kidscreen-27/

(8) Line 122. Was the KIDMED 1.0 or 2.0 used?

(9) Line 124, is "-4" correct?

(10) The materials and methods describe several varibles to be measured. Should they be also described and summarized in a table as appendix? (maybe it is not necessary)

(11)  Table 1. Add "interquartile range" words in the footnote.

(12) Could you please explain why Table 1 only shows the information at the follow-up time? Should the clinical data be recorded at diagnosis as well?

(13) Line 176. Please describe the content of the EuroNet-PHL-C1 therapy (First international Inter-Group Study for classical Hodgkin`s Lymphoma in Children and Adolescents).

(14) Line 190, regarding IDF criteria, did you use all the variables of the IDF calculator?

(15) In Table 3. Is the max value of the score 100?

(16) Line 316. Regarding "Our study demonstrates that obesity, high blood pressure and low HDL are present early in the survivorship of patients treated for Hodgkin lymphoma."

16.1. In the tables, could you please add the normal reference values?

16.2. In comparison to normal population of same age and sex, are the patients different? I.e. is obesity higher in treated patients than non-Hodgkin lymphoma children/young adults?

(17) In this study, most of the variables were measured at followup time. Should the variables also be measured at diagnosis? Therefore, the statisical analysis could include pre and post treatment data (for example, Wilcoxon (paired) signed-rank test).

(18) In the abstract, please add the number of cases of the series. Please also add the data as *** / *** (*** %). And P values when necessary.

Author Response

Dear Reviewer,

Thank you for giving us the opportunity to submit a revised draft of the manuscript “Metabolic sequelae and quality of life in early post-treatment period in adolescents with Hodgkin lymphoma” for publication in the Journal of Clinical Medicine. 

We appreciate the time and effort that you dedicated to providing feedback on our manuscript and are grateful for the insightful comments and valuable improvements to our paper.

We have incorporated most of the suggestions made by you and fellow reviewer, and changes are highlighted blue within the manuscript.

Please see below for a point-by-point response to your comments and concerns.

Comment 0: This article evaluated several metabolic syndrome variables and the quality of life of adolescents who had received treatment for Hodgkin lymphoma. The limitation is the number of cases (this is acknowledged at the end of the manuscript). Additionally, some variables were not measured before treatment. Therefore, I am not sure if "metabolic sequelae" should be used (authors know best). The article is easy to read and understand.

Response 0: With the population of about 3 million, Croatian scientific medical community is constantly battling with ‘small sample size’ as a limitation in medical research. According to the Croatian Cancer Registry, 99 citizens were diagnosed with HL in 2021., of whom 17 aged 0-19 years, treated at four different pediatric oncology centers.

The primary objective of our study was to determine metabolic sequelae in early post-treatment period in adolescents treated for HL. At the start of the treatment, only glucose, blood pressure and BMI were evaluated. Other MetS indices such as triglycerides and cholesterol were determined in minority of patients at the time of diagnosis. This is because majority of patients had normal values of fasting glucose as well as BMI z-scores, therefore, lipid disorders, according to our vast clinical experience,  were not anticipated. Nevertheless, we are aware that this is limitation of our study. However, we’ve decided that we will keep the “metabolic sequelae” in the title of our manuscript.

Comment 1: Lines 35-40. The authors could add some information (if they agree with this comment) regarding the different definitions of metabolic syndrome, which include NCEP ATP III 2005*, IDF 2009, EGIR 1999, WHO 1999, and AACE 2003.

Response 1: The first reference in the first paragraph of the manuscript is by Huang PL, regarding different definitions of metabolic syndrome, incorporating references suggested by the reviewer, so not to extend introductory part any further, we have decided to mention compilation definition only.

Comment 2: Line 42. Regarding "globally accepted definition of MetS in children is still lacking". They authors could mention the definitions of metabolic syndrome in children and adolescents, including the Modified ATP III, IDF (10 to 16 years), and NHANES III.

Response 2: IDF definition for children and adolescents has been used in the study and explained in detail in Methods (reference 3). Also, an elaborate sentence on main variables of the definition has been added to the second paragraph of the Introduction.

Comment 3: Line 58. Regarding "Every component of the MetS per se, quite understandably, appears to be even more prevalent in adult survivors of pediatric cancer". Could you please add the reference?

Response 3: References have been added.

Comment 4: In the Introduction, the authors could make a brief overview of the clinicopathological characteristics of Hodgkin lymphoma in children and adolescents.

Useful sentences are as follows:

(4.1) "Hodgkin lуmрhοmа (HԼ, formerly called Hodgkin's disease) is a malignant lуmphоma that accounts for approximately 7 percent of ϲhilԁhοоd cancers and 1 percent of ϲhildhοοԁ ϲаոсer deaths in the United States"

"The incidence of НԼ in ϲhilԁhοοd varies by age such that HL is exceedingly rare in infants, but is the most common ϲhilԁhοοd ϲаnсer in the 15- to 19-year-old age group."

"The overall incidence of HL in аԁοlеѕсеոts is greater in females than in males (male to female ratio 0.8); however, boys under age 15 have higher incidence of ΗԼ, with up to fivefold higher incidence of HԼ in boys than girls less than five years of age"

(4.2) Predisposing factors:

(a) Immunodeficiency (fore example, ataxia telangiectasia, chromosomal breakage syndromes, and the autoimmune lymphoproliferative syndrome.

(b) Familiar (familial HL has been estimated to represent 4.5 percent of all cases of HL. This familial association may include shared environmental factors, exposure to viruses, and genetic influences, including inherited immunodeficiency states).

(4.3) Histologic classification

(4.4) Clinical presentation

(4.5) Evaluation, diagnosis, and differential diagnosis

(4.6) Staging

(4.7) Treatment

Response 4: The reviewer’s suggestion has been accepted, as well as recommended reference (uptodate).

Comment 5: The authors may briefly describe the late complications of the treatment. Classically, the complications were impaired grоwth of soft tissue and bones, thyroid dysfunction, gonadal dysfunction, cardiopulmonary toxicity, second malignancies, and functional impairment and reduced overall general health.

Response 5: A list of the most common late complications has been added.

Comment 6: What kind of equipment was used to measure the bioelectrical impedance?

Response 6: Dear reviewer, thank you for your comment and question. Measurements were made using Maltron BF906 (Maltron International Ltd., Rayleigh, Essex, UK). This is a tetra-polar device, with an impedance of 200–1000 Ω, precision of ±4 Ω, and a frequency of 50 kHz. Patients were fasting for at least 4 h before testing and the measurement was performed with the participant lying in the supine position. We added this information to the Method section of the manuscript.

Comment 7: Line 118. Please add the website links:

https://www.who.int/tools/whoqol/whoqol-bref

https://www.kidscreen.org/english/questionnaires/kidscreen-27/

Response 7: Thank you for your comment. Website links have been added as references to the Manuscript.

Comment 8: Line 122. Was the KIDMED 1.0 or 2.0 used?

Response 8: Thank you for your question. The newest version of the KIDMED has been used. The reference for the exact version is indicated in the Manuscript.

Comment 9: Line 124, is "-4" correct?

Response 9: Yes, this is correct. This is because the KIDMED questionnaire has 4 questions which are scored with -1 in case the statement is true, and therefore, -4 is the lowest possible score.

Comment 10: The materials and methods describe several varibles to be measured. Should they be also described and summarized in a table as appendix? (maybe it is not necessary)

Response 10: Thank you for your comment and suggestion. We believe all relevant data described either in Tables or in the text form. We would like to keep the tables concise so it is easily followed by the reader. Therefore, we do not think it is necessary to add extra table in appendix.

Comment 11: Table 1. Add "interquartile range" words in the footnote.

Response 11: Thank you for your observation! This has been added as suggested.

Comment 12: Could you please explain why Table 1 only shows the information at the follow-up time? Should the clinical data be recorded at diagnosis as well?

Response 12: Thank you for your comment. As described in the Methods, some clinical data was available only at the follow- up, since the clinical data for the diagnosis was retrospectively extracted from the patient’s medical record. Unfortunately, for some of the variables we did not have the data at diagnosis.

Comment 13: Line 176. Please describe the content of the EuroNet-PHL-C1 therapy (First international Inter-Group Study for classical Hodgkin`s Lymphoma in Children and Adolescents)

Response 13: Details on the study have been included in the Results section.

Comment 14: Line 190, regarding IDF criteria, did you use all the variables of the IDF calculator?

Response 14: Thank you for your comment. Yes, all proposed variables were used.

Comment 15: In Table 3. Is the max value of the score 100?

Response 15: Thank you for your question.  There are no total scores for the KIDSCREEN-27. The three versions of the KIDSCREEN (KIDSCREEN-10, -27 & -52) were developed on the basis of the item response theory (Rasch model) where T-scores are calculated. This has some methodological advantages, such as the fact that the calculated T-scores of the tested children and adolescents are at interval scale level and thus parametric statistical procedures can be applied. More information on this you can find on the webpage: https://www.kidscreen.org/english/analysis/evaluation-by-hand/

Comment 16: Line 316. Regarding "Our study demonstrates that obesity, high blood pressure and low HDL are present early in the survivorship of patients treated for Hodgkin lymphoma."

16.1. In the tables, could you please add the normal reference values?

16.2. In comparison to normal population of same age and sex, are the patients different? I.e. is obesity higher in treated patients than non-Hodgkin lymphoma children/young adults?

Response 16.1: Thank you for your suggestion. Available reference ranges were added to the Table 2.

Response 16.2: Thank you for your comment. It seems that compared to Croatian population, prevalence of obesity is higher in survivors of HL. However, since we did not have a control group, we were only able to test this statistically.

Comment 17: In this study, most of the variables were measured at followup time. Should the variables also be measured at diagnosis? Therefore, the statisical analysis could include pre and post treatment data (for example, Wilcoxon (paired) signed-rank test).

Response 17: Thank you for your comment. As explained in Response 12, not all variables were measured at the diagnosis, and therefore, we were not able to compare some of the variables.

Comment 18: In the abstract, please add the number of cases of the series. Please also add the data as *** / *** (*** %). And P values when necessary.

Response 18: Thank you for your comment. Unfortunately, due to very limited number of words allowed in Abstract, we were able to add only the most important data. We hope this is enough to give the reader the most important information and to encourage the reader to read the whole manuscript.

Round 2

Reviewer 1 Report

Comments and Suggestions for Authors

It could be ideal as a letter to the editor in a journal of local relevance. The authors should either focus only on QoL or conduct a prospective study. You cannot infer any of the metabolic disturbances that occurred were due to HSCT since you do not have pre-HSCT data. A small study sample is not an argument to explain the methodological flaws. You can contact the patients and ask for pre-HSCT lipid profile values as they might know these values. You did not make any effort to address your limitations.